# Fluid Lubrication and Cooling Effects in Diamond Grinding of Human Iliac Bone

**DOI:** 10.3390/medicina57010071

**Published:** 2021-01-14

**Authors:** Yoshihiro Kitahama, Hiroo Shizuka, Ritsu Kimura, Tomo Suzuki, Yukoh Ohara, Hideaki Miyake, Katsuhiko Sakai

**Affiliations:** 1Spine Center, Omaezaki Municipal Hospital, Shizuoka 437-1696, Japan; tomos.pc@gmail.com; 2Medical Photonics Research Center, Hamamatsu University School of Medicine, Hamamatsu 431-3192, Japan; hmiyake@hama-med.ac.jp; 3Department of Mechanical Engineering, Faculty of Engineering, Shizuoka University, Hamamatsu 422-8529, Japan; shizuka.hiroo@shizuoka.ac.jp (H.S.); kimura.ritsu.15@shizuoka.ac.jp (R.K.); sakai.katsuhiko@shizuoka.ac.jp (K.S.); 4Department of Neurosurgery, Juntendo University School of Medicine, Tokyo 113-8421, Japan; yukoh@juntendo.ac.jp

**Keywords:** spine surgery, bone cutting, grinding, fluid lubrication, cooling effect

## Abstract

*Background and Objectives:* Although there have been research on bone cutting, there have been few research on bone grinding. This study reports the measurement results of the experimental system that simulated partial laminectomy in microscopic spine surgery. The purpose of this study was to examine the fluid lubrication and cooling in bone grinding, histological characteristics of workpieces, and differences in grinding between manual and milling machines. *Materials and Methods:* Thiel-fixed human iliac bones were used as workpieces. A neurosurgical microdrill was used as a drill system. The workpieces were fixed to a 4-component piezo-electric dynamometer and fixtures, which was used to measure the triaxial power during bone grinding. Grinding tasks were performed by manual activity and a small milling machine with or without water. *Results:* In bone grinding with 4-mm diameter diamond burs and water, reduction in the number of sudden increases in grinding resistance and cooling effect of over 100 °C were confirmed. *Conclusion:* Manual grinding may enable the control of the grinding speed and cutting depth while giving top priority to uniform torque on the work piece applied by tools. Observing the drill tip using a triaxial dynamometer in the quantification of surgery may provide useful data for the development of safety mechanisms to prevent a sudden deviation of the drill tip.

## 1. Introduction

Laminectomy in spinal surgery can be compared to some tasks in the industrial field. For example, cutting with diamond burs is similar to grinding, whereas cutting with steel burs is similar to cutting. The main effects of fluid on grinding and cutting include lubrication, cooling, and weld protection. Quantification and visualization of surgery that improve technology and have an educational effect are essential for the development of surgery robots [1]. In the field of orthopedics, there have been observational studies on bone cutting with an end mill [2,3,4,5,6,7,8,9]. However, there have thus far been few observational studies of bone grinding with diamond burs [10]. Our goal is the quantification and visualization of spine surgery [11,12,13,14,15,16,17,18,19,20]. One of our projects to achieve the goal is the environmental quantification of laminectomy using human iliac bones as workpieces. This study examined the effect of fluid applied by an assistant during spinal surgery on bone grinding. The purpose of this study was to examine the lubrication and cooling effects using a dynamometer and thermometer in an environment simulating laminectomy in microscopic spine surgery (dry and wet environments).

## 2. Materials and Methods

The measurement was performed in an environment simulating microscopic spine surgery. Thiel-fixed human iliac bones (a 93-year-old man, body-weight 50 kg, there was no history of osteoporotic fracture in his life) were used as workpieces. A neurosurgical microdrill (Primado 2, NSK-Nakanishi Inc., Kanuma, Japan) was used as a drill system. The rotation rate was set to 64,000 rpm. The neck length of the handpiece was 85 mm. A diamond electrodeposition grinding stone (PDS-2CDL-40; tip ball diameter, 4.0 mm: NSK-Nakanishi Inc., Kanuma, Japan) with a grain size 181–271 μm was used as a grinding tool. The workpieces were fixed to a 4-component piezo-electric Kistler dynamometer 9272 (Kistler Instrumente AG, Winterthur, Switzerland) using a rotary multi-vise (Model 2500-01: Dremel, Prospect, IL, USA) and fixtures, which was used to measure the triaxial power during bone grinding. Grinding tasks were performed by manual activity and a small milling machine (KitMill CL100; Originalmind Inc., Okaya, Japan). The conditions for manual grinding were as follows: the tilt of the handpiece, about 60°; cutting depth, about 2 mm; and feed rate, about 50–60 mm/min. We recorded the grinding distance at each measurement (L). The time required for the grinding were calculated by subtracting the time of the resistance value from the dynamometer record (T), and we found the speed (V) of each measurement (V = L/T: mm/s). The average value was calculated and used as the average speed during manual grinding. The conditions for grinding with a milling machine were as follows: the tilt of the handpiece, 0°; cutting depth, 2 mm; and feed rate, 55 mm/min. The temperature of the tool tip and its surroundings was measured using a thermal imaging camera (CHINO CPA-0130, Tokyo, Japan). In each task (grinding of cortical and cancellous bones), measurements in a dry environment (Dry), without applying perfusion fluid (tap water), and wet environment (Wet) were recorded with a dynamometer and a thermometer.

The contents of the six challenges (experimenter, osseous tissues, grinding direction, and liquid environments) were as follows: ① manual, cortical bone, Y (−), Dry (a) and Wet (b); ② manual, cancellous bone, Y (−), Dry (a) and Wet (b); ③ milling machine, cortical bones, X (+), Wet; and ④ milling machine, cancellous bone—cortical bone—air—cortical bone—cancellous bone, X (+)–Y (+)–X (−), Wet. For each 30 s task, the number of changes in sudden resistance peaks (spikes) and temperature were compared. We observed a high power with Wet condition with milling machine, so we predicted that the drill might be damaged, so we stopped measuring Dry condition with the milling machine. The definition of spike is as follows. When there was a change of 5 N or more from the average value of the dynamometer at the time of measurement and a change of 5 N or more from the measured value immediately before and after, it was counted as one Spike. Each procedure repeat three times and we selected most stable measurement, except with ② Wet. ② Wet was the record of slippage of the grinding tip.

We would like to state that the protocol for the research project has been approved by a suitably constituted Institutional Review Board (No. 19-193, dates of approval: 22 August 2019) within which the work was undertaken and that it conforms to the provisions of the Declaration of Helsinki in 1995 (as revised in Edinburgh 2000).

## 3. Results

The numbers of spikes were as follows: ① Dry, Cortical-X 23, Y 10, Z 21, ① Wet, Cortical-X 0, Y 0, Z 0, ② Dry, Cancellous-X 0, Y 0, Z 5, ② Wet, Cancellous-X 0, Y 1, Z 11 (Figure 1 and Figure 2) ③ Wet, Cortical-X 92, Y 10, Z 19, and ④ Wet, Cancellous and Cortical-X 21, Y 25, Z 21 (Figure 1 and Figure 3). The components of each direction of the measurement result were recorded by three colors. The correspondence between the directions and the colors was as follows (x, y, z): (green, red, violet). The temperatures in ① were as follows: Dry, >270 °C, Wet, 61 °C (Figure 4). In ① and ② (manual grinding), each of the triaxial components were within −5.5~7.5 N, and the peak of the spikes were within −12.5~15.0 N. In ③ and ④ (grinding with a milling machine), in some cases, each component of the three axes were within −10~11 N, and the peak of the spikes were −80~80 N (Table 1).

## 4. Discussion

When the direction of travel was Y, assume that the components of the three axes, X, Y, and Z reflect torque stability, stable feed speed, and stable cutting depth, respectively, like in the manual ① and ② series. In ③ and ④, machine grinding travel X, X: feed, Y: torque, Z: depth were reflected. Furthermore, each measurement result was examined separately as follows: (1) differences due to the perfusion fluid environment; (2) differences due to the tissue; and (3) differences between manual and milling machines.

(1)Differences due to the perfusion fluid environments

Each of the measurements ① and ② performed in the same grinding environment (Dry and Wet) was an experimental system in which it was easy to compare the effects of fluid. In ①, ②, and ③, spikes almost disappeared in the wet environment. This result may be due to water lubrication. A prominent cooling effect (a decrease in temperature over 100 °C in the wet environment) was observed. Because 50% of irreversible changes in the nerve root are reported to occur at 44 °C and 30 min [21], intraoperative application of water by an assistant may be essential to protect the nerves.

As a treatment that uses thermal energy, high-frequency hyperthermic coagulation therapy is used to treat pain such as chronic low back pain [22,23,24]. C5 palsy after cervical spine surgery is well-known as a complication as a neurological disorder caused by thermal damage associated with drilling [25,26,27]. Unmyelinated fibers were immediately destroyed at 58 °C [28]. It is important to prevent complications that the bone cutting procedure near the affected nerve root is completed under water perfusion in the shortest time. Percutaneous endoscopic bone cutting is performed with a 3.5 mm diameter diamond burs under continuous perfusion of saline. It is considered to be a method that enables fine work in an environment where thermal damage is not to occur and may contribute to improving surgical outcomes [29]. Under the microscope, a mist irrigation system has been developed, and the operator’s spraying of water and the cooling action of the nitrogen gas injection contribute to improving the safety in addition to cleaning the operative field [30].

(2)Difference due to the tissue

In ①, ③, and ④, cortical bones were ground, whereas in ② and ④, cancellous bones were ground. In manual grinding, spikes were often observed in cancellous bones, whereas in grinding with a milling machine, spikes were frequently observed in both cortical and cancellous bones. The spikes that were frequently observed in the *z*-axis during manual grinding of cancellous bones may be due to an attempt to stabilize the cutting depth while considering the histologically heterogeneous properties of cancellous bones.

Generally, during laminectomy, cancellous bone resection is an intermediate process until reaching the deep cortex, and is not a site requiring stable work. The deep cortex is in contact with the nerve root and dural canal via the ligamentum flavum, so a stable and fine procedure is required. Robotization needs to be able to safely perform deep cortical cutting. It was difficult to detect the loss of resistance due to the tilt angle between the deep cortex and the bur during mechanical bone grinding [10,31]. We observed the phenomenon that the tip of the bur was flipped by the manual operation (②w). Before the rapid deviation of the drill tip, when the pressure changes of the three axes were observed for each component, an unstable pressure increase of components other than torque related to Feed and Depth were observed as a sign. In the case of manual operation, the penetration of the bone cortex is sensed with the fingertips to prevent the nerve root injury by the tip. The development of a master-slave robot with a tip pressure change feedback mechanism will contribute to safe treatment of deep cortex.

Each spinal surgeon has a technique for penetrating cortical bone. The eggshell procedure is a famous method of decompressing nerve tissue since the microscopic era [32]. While applying the equatorial plane of the bur tip to the grinding surface, thin the cortex and apply pressure as little as possible to the contact area, and see the blood vessels around the nerve through the thinned cortical bone, and fluctuation of CSF pressure. There are many surgeons who use Kerrison forceps to complete the final decompression by avoiding the sudden involvement and sudden change in grinding depth. Digitization of these technologies will contribute to future software development for robot control.

For robot development, automation of the bone cutting operation by the eggshell procedure is one of the goals. The situation in which it is possible to control instruments in 1nm units and the development of various sensors will help our works. It is possible to develop the new powerful arm for the Kerrison forceps in with the extension line of the existing robot arm. It is necessary to grind until the cortical bone can be thinned by the Kerrison forceps. Therefore, we can never avoid the development of bone grinding arms in the development of spinal surgery robots. This research advances the development of a safe cortical bone treatment technique that never damages nerves as a program by linking tissue characteristics with precise position information and linking information obtained from sensors such as drill vibrations with position information [33,34,35,36,37,38,39,40].

(3)Difference between manual and milling machines

In ① and ②, grinding was performed by manual activity, whereas in ③ and ④, grinding was performed by a milling machine. Manual grinding showed a stable torque load, whereas grinding with a milling machine had the highest load in the direction of feed. The primary contributor to the difference may be tool tilt. The tool rotation speed is highest in the equatorial plane (when the radius is largest) and lowest at the tip. Because the rotational speed is reflected in cutting efficiency, dynamometer values are lower near the equator and higher at the tip. When the tools with low cutting efficiency are advanced at a constant speed, the torque is large. The differences between manual and milling machines in triaxial waveform components suggested that manual grinding adjusts the feed speed and cutting depth to maintain torque resistance.

In manual grinding (② Wet), measurement ended due to tip deviation, in which the *z*-axis load suddenly became 0. Prior to that phenomenon, the dynamometer detected an increase in the Z-component. The findings suggest the possibility of developing a system that predicts tip deviation in advance and activates the safety mechanism through data accumulation.

In order to safely perform a drill grinding procedure via a master-slave type robot arm, in addition to high-definition movie image information of the surgical field, precise position information of the drill tip and tactile information from a pressure sensor are required. Real-time position information needs to reflect changes in breathability and subduction due to the load on the drill tip and outer cylinder tip [10]. Surgeons consider that tactile information is more important than high-definition visual data in spinal surgery drill performance. Even if a high-definition visual data of 4K or more under water irrigation is obtained endoscopically, the turbidity of the surgical field due to grinding pieces and water bubbles is unavoidable with the current technology. The tactile information helps fine work for the operator. The key to development is how to subtract the noise information, such as the vibration of the motor and the contact between the tip of the outer cylinder and the bone wall, from the pressure information of the slave arm that holds the drill and return it to the master operation site. At present, it is in a state where the surgical results can be improved more efficiently by instructing the young surgeon than by developing a robot. When this is reversed, robot spine surgery will become popular.

### 4.1. Limitations of this Study

Data comparisons require caution because the following factors may have affected the measurement results. The tool tilt in manual grinding was not equal to that in grinding with a milling machine. Because the temperature measurement was performed using the non-contact infrared measurement method, the temperature measurements of the tool tip and workpieces may be more inaccurate than the actual measurements by the contact sensors. Since saline was applied to the surgical field during surgery, lubrication and cooling effects may differ between this experiment using tap water. Because the spine has a complex shape with more irregularities than iliac bones, the data obtained from this study in an experimental environment may be different from measurement data obtained during spine surgery.

### 4.2. Challenges and Prospects for Future Studies

In this study, it is necessary to further study the difference in measured values due to the experience of the operator and the difference in measured values due to changes in the conditions of milling machine. A bone cutting system using mist irrigation (MI), which is similar to cutting under minimum quantity lubrication (MQL) in the field of engineering, is commonly used in microscopic spine surgery. Observational research on grinding using MI, which is similar to MQL cutting, is essential for the development of future surgical robots. The results of this study suggest that percutaneous endoscopic spine surgery, which is irrigation surgery, is a favorable environment for bone cutting. Future studies should examine the effect of measurements in an environment that imitates percutaneous endoscopic spine surgery.

## 5. Conclusions

In our observational study using a triaxial dynamometer and thermometer, which simulated laminectomy in microscopic spine surgery, we confirmed that fluid lubrication and cooling are useful in the prevention of irreversible changes in nerve tissues near the spine.

The manual grinding gives priority to a stable torque, whereas grinding with a milling machine gives priority to a stable feed and depth.

Observing the drill tip using a triaxial dynamometer in the quantification of surgery may provide useful data for the development of safety mechanisms to prevent a sudden deviation of the drill tip.

## Figures and Tables

**Figure 1 medicina-57-00071-f001:**
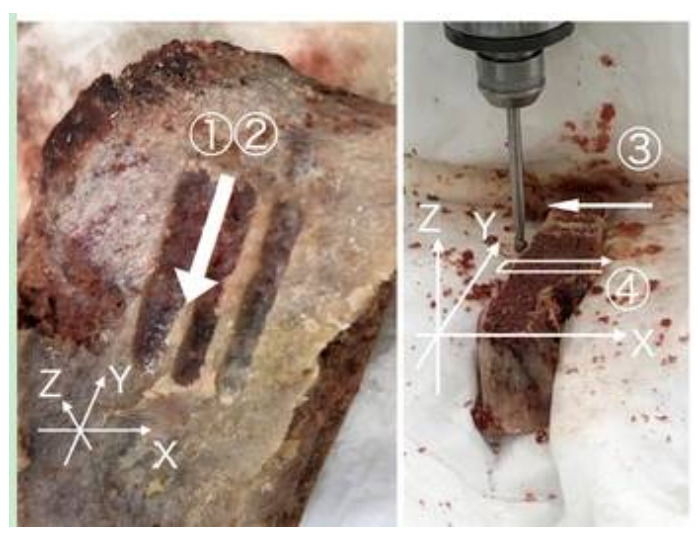
Relationship between the material and the grinding procedure. The left figure is a photograph after manual grinding of the cortex. The photo on the right is the machine procedure. Every series was repeated three times and recorded in the data logger.

**Figure 2 medicina-57-00071-f002:**
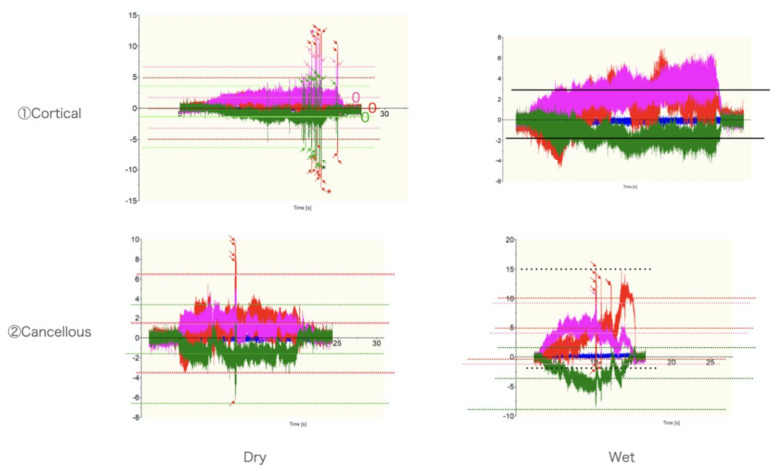
Typical records for manual grinding. The data were compared for differences in the tissues (cortical and cancellous) and the presence of water (dry and wet). The fluctuation range of the dynamometer and the number of rapid fluctuations (spikes) in the measured value were observed for each trial.

**Figure 3 medicina-57-00071-f003:**
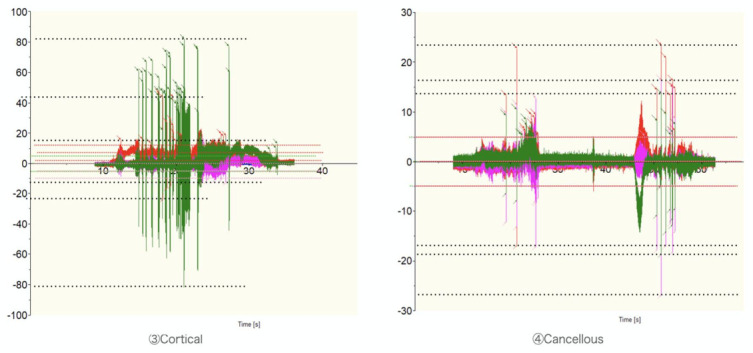
Typical records for machine grinding. The data were compared for differences in the tissues (cortical and cancellous) in the wet atmosphere. The fluctuation range of the dynamometer and the number of rapid fluctuations (spikes) in the measured value were observed for each trial.

**Figure 4 medicina-57-00071-f004:**
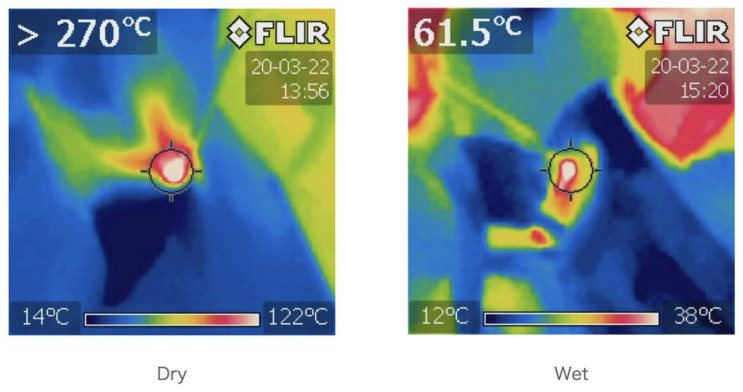
The records of the cooling effect of water when manually grinding cortical bone. Dry atmosphere on the left and wet atmosphere on the right. Water had a cooling effect of more than 100 °C in human bone grinding.

**Table 1 medicina-57-00071-t001:** The grinding force records of six procedures. When there was a change of 5 N or more from the average value of the dynamometer at the time of measurement and a change of 5 N or more from the measured value immediately before and after, it was counted as one spike.

Series	Tissue	Atmosphere	Feed	Direction	Grinding Force (Base ± Amplitude, N)			Spike (Frequency & Force, N)		
					x	y	z	x	y	z
1d	Cortical	Dry	Manual	y(−)	−1.5 ±1.5	−2.0 ±2.0	0 ±1.0	23 −10~5	10 −1~11	21 −12~13
1w	Cortical	Wet	Manual	y(−)	−2.0 ±2.0	−3.0 ±3.0	−3.0 ±4.0	0	0	0
2d	Cancellous	Dry	Manual	y(−)	−1.5 ±1.0	1.5 ±1.5	1.5 ±2.5	0	0	5 −6~10
2w	Cancellous	Wet	Manual	y(−)	−3.5 ±2.0	4.0 ±1.5	5.0 ±2.5	0	1 11	11 −2~15
3	Cortical	Wet	Machine	x(−)	0 ±5.0	−5.0 ±5.0	−8.0 ±3.0	92 −80~80	10 −12~15	19 −23~43
4	Cancellous Cortical	Wet	Machine	x(−)y(5)x(+)	0 ±3.0	0 ±3.0	0 ±3.0	21 −18~13	25 −27~17	21 −17~24

## Data Availability

The data that support the findings of this study are available from the corresponding author, Yoshihiro Kitahama, upon reasonable request.

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
