# Peer review of "Fluid Lubrication and Cooling Effects in Diamond Grinding of Human Iliac Bone"

_medicina, 2021, doi:10.3390/medicina57010071_

Round 1

Reviewer 1 Report

It's an interesting study to evaluate simulated surgical environment during bone drilling in respect to heat and stability with fluid lubrication 

Author Response

Dear Reviewers, 

Thank you for your comments. We have revised our manuscript along the lines specified by the reviewer. The changes are as follows:

  1. Figure 2: We posted Figure 2 as Figure 3. We could notice it ,as suggested by the reviewer. We try to post right numbers of Figure 2 and 3.

2.   We have added the following six items, as suggested by the reviewer. 

 â‘  How did you verify the speed of manual grinding?

We added to the Material and Methods. “We recorded the grinding distance at each measurement (L). The time required for the grinding were calculated by subtracting the time of the resistance value from the dynamometer record (T). And we found the speed (V) of each measurement (V = L / T: mm / sec). The average value was calculated and used as the average speed during manual grinding. “

â‘¡ Why did not you compare the dry and wet condition when cutting with milling machine?

Thank you for your appropriate advice. In the case of milling machine, the load were heavy under Wet conditions, so we afraid the drill could be damaged under Dry condition. In the future, even if the drill is damaged, when we can ready to the same drill, we plan to carry it out. 

We added to the method, "We observed a high power with Wet condition with milling machine, so we predicted that the drill might be damaged, so we stopped measuring Dry condition with milling machine."

â‘¢ Section of the method is not completed.

Thank you for your advice. I made a correction.

â‘£ Figure 2 shows that cortical and cancellous tissue grinding by milling machine accord to the number written below the figures, but the caption shows it is manual grinding and machine grinding. Please indicate the accurate explanation. What do the red and green and violet color indicate?

Thank you for reviewing the data in detail. We checked again uploaded manuscript and noticed your comment’s meaning. Figure 2 is our Figure 3. We will upload our Figure 2 again. Thank you. 

All measurements in our data of Figure 2 are manual, and the data of Figure 3 are milling machine.

And I added to the result, "The components of each direction of the measurement result were recorded by three colors. The correspondence between the directions and the colors was as follows (x, y, z): (green, red, violet)."

⑤ To compare the manual and milling machine, the specimens should be equal (left and right same part of the pelvic for example). There is no data about the cadaver (age, sex osteoporosis, how many?).

You are right. When material were homogenous substance like metal for grinding, uniform data can be obtained. But when heterogeneous substance like human specimens, it is difficult to prepare the good conditions for the material. We prepared to measure the bone density, but we could not prepare the environment in the training room. Ethical members in our institute(Hamamatsu University School of Medicine) considered our measurement should complete in the training room. We could not carry the specimens to the Computer Tomography room last year. In the future, we plan to improve the environment, assuming that we will consider the difference in measurement results due to bone density.

In addition to the data shown, similar data were collected from 90-year-old two women, and it was confirmed that they have the same tendency.

From the above, I just added the following to the method.

“Thiel-fixed human iliac bone (a 93-year-old man, body-weight 50 kg, There was no history of osteoporotic fracture in his life) was used as workpiece.”

â‘¥ The manual method should be performed by several surgeons, and the clinical experience of the surgeons should be described.

In the future, we will request the cooperation of young surgeons and young researchers to examine the difference in measured values ​​due to surgical experience.

The following text has been added to the beginning of the Future studies paragraph.

"In this study, it is necessary to further study the difference in measured values ​​due to the experience of the operator and the difference in measured values ​​due to changes in the conditions of milling machine."

 Through this research, we will continue to contribute to the development of medicine and collaboration with the engineering field. We strive not only to contribute to academia, but also to reach the social implementation of technology. We look forward to your continued support. Thank you again.

Sincerely,

Yoshihiro Kitahama, MD

Cooperated Major of Medical Photonics, Hamamatsu University School of Medicine

Reviewer 2 Report

The application of robotics to spinal surgery is in its infancy.

This study has the potential to make a significant contribution to the safe application of fully robotic spine surgery.  As companies develop their drill system it will be important to them to know the drip rate of water on the drill and automatic "stops" to the drill.  

This article can help answer these questions however in its current draft it is hard to get the information.  It would be very helpful if the authors can highlight these important points.

Author Response

(The authors gave the same response as above.)

Reviewer 3 Report

The authors compared the characteristics of grinding between manual and robot grinding in several conditions. 

Although the authors concluded that the irrigation with water will prevent the nerve or tissue damage near the spine, the experiment in this study showed only a decrease in temperature with irrigation. if the authors want to show the clinical significance, surrounding temperature should be measured continuously with changing the amount and the temperature of the irrigation water.

Another conclusion of the authors is that the manual grinding gives a stable torque compared to a milling machine. This fact is natural in a setting where the drill is moved at a constant speed. If the milling machine is moved at a constant speed, it is natural that the load will increase depending on the hardness of the tissue. On the other hand, it is natural for humans to have a constant load because the speed cannot be constant when cutting tissue with different hardness.

Other comments:

1. How did you verify the speed of manual grinding?

2. Why did not you compare the dry and wet condition when cutting with milling machine?

3. Section of the method is not completed.

4. Figure 2 shows that cortical and cancellous tissue grinding by milling machine accord to the number written below the figures, but the caption shows it is manual grinding and machine grinding. Please indicate the accurate explanation. What do the red and green and violet color indicate?

5. To compare the manual and milling machine, the specimens should be equal (left and right  same part of the pelvic for example). There is no data about the cadaver (age, sex osteoporosis, how many ?). The manual 

6. The manual method should be performed by several surgeons, and the clinical experience of the surgeons should be described.

Author Response

(The authors gave the same response as above.)

Round 2

Reviewer 2 Report

Authors,

Thank you for your contribution to the literature.  Future research and applictions will help make robotic spine surgery safe.  

Reviewer 3 Report

The authors have answered the questions adequately.